# Uncovering the Interrelation between Metabolite Profiles and Bioactivity of In Vitro- and Wild-Grown Catmint (*Nepeta nuda* L.)

**DOI:** 10.3390/metabo13101099

**Published:** 2023-10-20

**Authors:** Anna Zaharieva, Krasimir Rusanov, Mila Rusanova, Momchil Paunov, Zhenya Yordanova, Desislava Mantovska, Ivanka Tsacheva, Detelina Petrova, Kiril Mishev, Petre I. Dobrev, Jozef Lacek, Roberta Filepová, Grigor Zehirov, Valya Vassileva, Danijela Mišić, Václav Motyka, Ganka Chaneva, Miroslava Zhiponova

**Affiliations:** 1Department of Plant Physiology, Faculty of Biology, Sofia University “St. Kliment Ohridski”, 1164 Sofia, Bulgaria; aazahariev@uni-sofia.bg (A.Z.); jiordanova@biofac.uni-sofia.bg (Z.Y.); d_mantovska@biofac.uni-sofia.bg (D.M.); detelina@biofac.uni-sofia.bg (D.P.); chaneva@biofac.uni-sofia.bg (G.C.); 2Department of Agrobiotechnology, Agrobioinstitute, Agricultural Academy, 1164 Sofia, Bulgaria; krusanov@abv.bg (K.R.);; 3Department of Biophysics and Radiobiology, Faculty of Biology, Sofia University, 1164 Sofia, Bulgaria; m_paunov@uni-sofia.bg; 4Department of Biochemistry, Faculty of Biology, Sofia University, 1164 Sofia, Bulgaria; itsacheva@biofac.uni-sofia.bg; 5Department of Molecular Biology and Genetics, Institute of Plant Physiology and Genetics, Bulgarian Academy of Sciences, 1113 Sofia, Bulgaria; mishev@bio21.bas.bg (K.M.); grig@bio21.bas.bg (G.Z.); valyavassileva@bio21.bas.bg (V.V.); 6Laboratory of Hormonal Regulations in Plants, Institute of Experimental Botany of the Czech Academy of Sciences, 165 02 Praha, Czech Republic; dobrev@ueb.cas.cz (P.I.D.); lacek@ueb.cas.cz (J.L.); filepova@ueb.cas.cz (R.F.); vmotyka@ueb.cas.cz (V.M.); 7Department of Plant Physiology, Institute for Biological Research “Siniša Stanković”, National Institute of the Republic of Serbia, University of Belgrade, 11060 Belgrade, Serbia; dmisic@ibiss.bg.ac.rs

**Keywords:** bioactivity, GC-MS, *Nepeta nuda*, phytohormones, metabolites, volatiles

## Abstract

*Nepeta nuda* L. is a medicinal plant enriched with secondary metabolites serving to attract pollinators and deter herbivores. Phenolics and iridoids of *N. nuda* have been extensively investigated because of their beneficial impacts on human health. This study explores the chemical profiles of in vitro shoots and wild-grown *N. nuda* plants (flowers and leaves) through metabolomic analysis utilizing gas chromatography and mass spectrometry (GC–MS). Initially, we examined the differences in the volatiles’ composition in in vitro-cultivated shoots comparing them with flowers and leaves from plants growing in natural environment. The characteristic iridoid 4a-α,7-β,7a-α-nepetalactone was highly represented in shoots of in vitro plants and in flowers of plants from nature populations, whereas most of the monoterpenes were abundant in leaves of wild-grown plants. The known in vitro biological activities encompassing antioxidant, antiviral, antibacterial potentials alongside the newly assessed anti-inflammatory effects exhibited consistent associations with the total content of phenolics, reducing sugars, and the identified metabolic profiles in polar (organic acids, amino acids, alcohols, sugars, phenolics) and non-polar (fatty acids, alkanes, sterols) fractions. Phytohormonal levels were also quantified to infer the regulatory pathways governing phytochemical production. The overall dataset highlighted compounds with the potential to contribute to *N. nuda* bioactivity.

## 1. Introduction

The phytochemical composition of medicinal plants varies depending on the geographical origin of the species, genotype variation, environmental conditions, developmental stage, and methods of handling the plant material [1,2,3]. The naked catmint, which is the common name of *Nepeta nuda* L., is a medicinal plant known to possess several chemotypes of the essential oil influenced by environmental conditions, such as temperature, precipitation, and insolation [4]. These chemotypes include the following: (1) with the iridoid monoterpene nepetalactone; (2) with the monoterpene 1,8-cineole/eucalyptol; (3) mixed types (nepetalactone, 1,8-cineole, and the sesquiterpene germacrene D); (4) nonspecific chemotypes. Dehydration of leaves has been pointed out as a negative regulator of nepetalactone metabolism in *Nepeta* species [5]. Additionally, an increase in altitude has been linked to the delayed accumulation of *N. nuda* essential oil [6]. Due to its composition of volatiles, *N. nuda* is characterized as a honey plant attracting primarily honeybees and representatives of the Felidae family as pollinators. Conversely, these compounds also contribute to the repellent activity against insects [7,8,9]. A recent report on *N*. *nuda* has demonstrated a relationship between the total phenolic quantity on the geographical origin [10]. Further comparisons between wild-grown plants and those cultivated in aseptic in vitro conditions have shown differing accumulation patterns of phenolic acids (aesculin, caffeic acid, chlorogenic acid, ferulic acid, protocatechuic acid, rosmarinic acid), flavonoids (apigenin, apigetrin, cirsimaritin, luteolin), and iridoid glycoside (epideoxyloganic acid). These variations corresponded to respective adjustment in antioxidant, antiviral, and antibacterial activities [10].

The study seeks to further unravel the putative role of different classes of molecules in the bioactive potential of *N. nuda* and their interdependence with the environment. Utilizing gas chromatography coupled to mass spectrometry (GC–MS), we conducted metabolomic analysis on different variants of *N. nuda* cultivated under variable conditions. In vitro grown-shoots, as well as flowers and leaves from ex vitro-adapted wild-grown plants were examined for their content of volatiles, and primary and secondary metabolites (Figure 1). By comparing the obtained chemical profiles with existing information on corresponding biological activities, we revealed the contribution of endogenous phytohormones in the context of growth conditions.

## 2. Materials and Methods

### 2.1. Plant Material

Whole shoots of in vitro-grown plants (5 weeks old), and individual flowers and leaves from ex vitro-adapted plants of *Nepeta nuda* L. ssp. *nuda* (at the phase of active blooming) were used in this study (Appendix A). Plant cultivation under in vitro conditions and subsequent adaptation to ex vitro conditions were carried out according to [10]. Thus, the in vitro- and wild-grown plants were genetically identical. During in vitro cultivation, plants were sterilely grown on MS medium supplemented with 3% sucrose and 0.7% agar [11], and controlled growth conditions: white light (80 μmol m^–2^ s^–1^ PAR, warm white fluorescent, MASTER TL-D Super 80 36W/840 1SL/25, Philips, Pila, Poland), a photoperiod of 16 h light/8 h dark, at 25 ± 1 °C, and moderate humidity ranging between 60 and 70%. Samples from ex vitro-adapted plants were collected in June.

### 2.2. GC-MS Analysis of Volatile Compounds

Extraction of volatile compounds was performed in a 20 mL GC headspace analysis vial. In each vial, 4 mL of ethyl acetate (Sigma, St. Louis, MI, USA) was added, and fresh plant material was completely immersed in the solvent. Vials were closed with a crimp and the samples were vortexed for 4 h. Subsequently, 200 mg of anhydrous sodium sulfate (Sigma) was added to each vial to remove residual water. After additional vortexing for 10 min, 1 mL of the resulting extract from each sample was transferred to a 2 mL vial for GC-MS analysis. Analysis of volatile compounds was performed on an Agilent 8890/5977BMSD/FID GC-MS system using an Agilent (Santa Clara, CA, USA) HP-INNOWax (PEG stationary phase) column with a length of 30 cm, an internal diameter of 250 μm and a coating thickness of 0.25 μm. From each sample, 1 μL was injected in a split less flow mode at the injector temperature of 250 °C. Helium 5.0 was used as carrier gas at a flow rate of 0.8 mL/min. The oven temperature program included an initial temperature of 65 °C, at a rate of 3 °C/min, ramping up to 220 °C and holding for 15 min. The mass detector temperature was set at 150 °C, and the transfer line temperature was 250 °C. Ionization was achieved through electron impact (EI) at 70 eV. To establish the retention index, a mixture of C10-C40 normal alkanes (Sigma) was used in conjunction with AMDIS ver. 2.73 (National Institute of Standards and Technology (NIST), Gaithersburg, MD, USA). The FID detector operated at a temperature of 300 °C. Compounds were identified by comparing the Kovach retention index and mass spectrum with those of standard substances and mass spectral data from the National Institute of Standards and Technology (NIST 08) libraries, the Robert Adams library [12] and pertinent literature data. MSD ChemStation software (F.01.03.2357, Agilent Technologies, Inc., Santa Clara, CA, USA) was used to integrate and determine the peak areas of individual volatile compounds. The amounts of individual components were expressed as a percentage of the total chromatogram area based on data from the FID detector. Compounds with content lower than 0.01% were presented as “traces”.

### 2.3. GC-MS Analysis of Primary and Secondary Metabolites

For each sample, 1 mL of methanol was added to 40 mg of dry plant material in a 2 mL GC analysis vial. Then, the following standard substances were added to each sample: 2 mg mL^−1^ nonadecanoic acid in methanol; 2 mg mL^−1^ ribitol in ddH_2_O; and 10 mg mL^−1^ 3,4,5-trimethoxyhydroxycinnamic acid in methanol. Next, extraction was performed using a thermomixer at 1000 rpm and a temperature of 70 °C for 30 min. After this extraction step, centrifugation at 3000 rpm for 10 min was carried out. To 500 μL of the resulting supernatant, 500 μL of chloroform was added, and the samples were vortexed at 1000 rpm for 10 min. This was followed by the addition of 500 μL ddH_2_O to each sample and vortexing at 1000 rpm for 5 min. Samples were centrifuged at 3000 rpm for 10 min to separate the polar fraction (upper phase) from the non-polar (lower phase). From the initial polar fraction, 250 μL were transferred to a clean 2 mL GC vial (PF1—polar fraction 1) for subsequent analysis of amino acids, sugars, and organic acids. From the remaining polar fraction, 250 μL were transferred to another 2 mL GC vial (PF2—polar fraction 2) for subsequent analysis of phenolic acids. Lastly, from the non-polar fraction (NPF), 400 μL was separated for subsequent analysis of fatty acids and sterols.

#### 2.3.1. Analysis of Amino Acids, Sugars, and Organic Acids

From the PF1 fraction, 250 μL was evaporated to complete dryness on a vacuum concentrator at 30 °С for about 2:30 h. To the resulting dry sample, 50 μL of pyridine (Sigma) and 50 μL of methoxyamine hydrochloride (20 mg mL^−1^ in pyridine) were added. After dissolving the samples, they were heated for 1 h at 80 °C in a thermomixer TMix (Analytik Jena, Jena, Germany) at 1000 rpm. Following this step, 50 μL of the silylating agent N,O-bis(trimethylsilyl)trifluoroacetamide (BSTFA, Sigma) was added, and another 1 h incubation at 80 °C in a thermomixer at 950 rpm was carried out. The prepared samples were analyzed on a system consisting of an Agilent GC 7890 gas chromatograph and an Agilent MD 5975C mass spectral detector. An Agilent HP-5ms ((5%-phenyl)-methylpolysiloxane) column was used for the separation of substances at the following temperature program: initiation at 80 °C, hold for 2 min, increase to 200 °C at a rate of 5 °C/min, hold for 2 min, increase to 300 °C at a rate of 10 °C/min, and hold steady for 10 min. One microliter of each sample was injected in a 10:1 flow split mode with the injector temperature of 250 °C. The mass detector operated at 200 °C and the transfer line temperature was maintained at 250 °C. Ionization was achieved through electron impact at 70 eV. To establish the retention index, a mixture of C10-C40 normal alkanes (Sigma) was employed, using AMDIS ver. 2.73 (NIST, USA). Identification of compounds involved the comparison of Kovach indices and mass spectral data from the National Institute of Standards and Technology (NIST 08) libraries, The Golm Metabolome Database, and the literature data.

#### 2.3.2. Analysis of Phenolic Acids

To 250 μL of the PF2 fraction, 500 μL of 1 M sodium hydroxide solution was added. The mixture was vortexed until the extract was completely dissolved and then allowed to stand in darkness for 18 h. Upon the completion of this incubation period, the samples were acidified to a pH of 2 using hydrochloric acid and incubated for 1 h at 90 °C at 950 rpm in a thermomixer. Following this step, three extractions were performed with 500 μL ethyl acetate for 30 min at 1000 rpm by vortexing. The resulting three fractions were pooled and transferred to a new vial. To dehydrate the solution, 50 mg of anhydrous sodium sulfate was added to the vial, followed by vortexing for 10 min at 1000 rpm. Next, the samples were evaporated to dryness using a vacuum concentrator at 30 °C for 60 min, after which 50 μL of pyridine and 50 μL of BSTFA (Sigma) were added. Subsequently, the samples were incubated for 1 h in a thermomixer at 80 °С at 1000 rpm, and the prepared samples were analyzed using GC/MS. The GC/MS analysis and compounds identification were as described in Section 2.3.1.

#### 2.3.3. Analysis of Fatty Acids, Sterols, and Hydrocarbons

The NPF fraction, comprising 400 μL, was evaporated to dryness using a vacuum evaporator at 30 °C for 15 min. Then, 500 μL of 2% sulfuric acid in methanol was added to the dry NPF material. The samples were incubated for 1 h at 90 °C/1000 rpm in a thermomixer. This was followed by cooling and triple extraction with 500 μL of n-hexane for 30 min at 1000 rpm. The hexane fractions were combined in a new vial followed by the addition of 50 mg of anhydrous sodium sulphate and vortexing for 10 min at 1000 rpm. The sample was completely dried using a vacuum evaporator, after which 50 μL of pyridine and 50 μL of BSTFA were added. The samples were then incubated for 1 h in a thermomixer at 80 °С at 1000 rpm, after which they were analyzed using GC/MS. The GC/MS analysis and compounds identification were as described in Section 2.3.1.

### 2.4. Anti-Inflammatory Activity

The procedure outlined by [13] was followed for conducting the microtiter hemolytic complement assay on *N. nuda* aqueous extracts. The extracts were prepared at 40 °C via maceration for 24 h and subjected to lyophilization. The dried extracts were dissolved in 3% dimethyl sulfoxide (DMSO) and diluted with barbitone buffered saline, pH 7.2, containing 0.15 mM Ca^2+^ (BBS). The assay was performed on 6% sheep erythrocyte (SE) suspension sensitized with rabbit polyclonal anti-SE serum (hemolysin) and guinea pig complement (BulBio, Sofia, Bulgaria). Preliminary titration of sera was performed to determine the dilution producing 50% hemolysis of target erythrocytes. The SE (25 μL/well) were sensitized via 30 min incubation with hemolysin (dilution 1:1600, 25 μL/well) at 37 °C. After that complement (125 μL/well of appropriate dilution), increasing amounts of the analyzed plant extracts (100 μL/well) were added and incubated for 1 h at 37 °C. Next, the microtiter plates were centrifuged at 1000× *g* for 5 min, and 200 μL of the supernatant from each well was transferred to new 96-well flat-bottom microtiter plates and the absorbance at 540 nm was measured spectrophotometrically. 

### 2.5. Anthocyanins and Reducing Sugars

The extraction of anthocyanin pigment was carried out by homogenizing dried plant material (10 mg mL^−1^) in acidified methanol (methanol:water:HCl, 79:20:1, *v*/*v*) according to [14]. Extracts were centrifuged for 10 min at 5000 rpm, and the absorbance was measured at 530 nm and 657 nm for each sample. Absorption was calculated using the formula [Corrected A530 nm = A530 nm^−1^/3 A657 nm].

To quantify the content of reducing sugars, the 3,5-dinitrosalicylic acid method was employed, as described by [15]. Dried plant material (10 mg mL^−1^) was homogenized in distilled water (dH_2_O) and boiled for 10 min. After centrifugation for 20 min/13,000 rpm, the supernatant was pipetted off for analysis. For each sample, 250 μL of supernatant and 250 μL incubation medium (0.04 M 3,5-dinitrosalicylic acid, 1 M K-Na-tartrate, 0.2 N NaOH), were mixed. In the control, instead of the supernatant, dH_2_O was added. The samples and the control were boiled for 5 min, and after cooling, 4 mL dH_2_O was placed. Following calibration against a control, the extinction at λ = 530 nm was measured spectrophotometrically. The content of reducing sugars was measured using a standard curve constructed using glucose as the reference standard.

### 2.6. Plant Hormone Profiling

The plant hormone profiling was carried out following the methods detailed in [16,17]. The plant material was frozen in liquid nitrogen (ca. 20 mg fresh weight/sample) and homogenized with zirconium oxide beads (5 mm diameter) in TissueLyser LT (Qiagen) for 2 min at 50 Hz. The extraction was carried out with 50% acetonitrile solution supplemented with a mixture of stable isotope-labelled internal standards (1 pmol/sample) [16]. The crude extracts were centrifuged, and the supernatants were subjected to solid phase extraction using Oasis HLB 96-well column plates (10 mg/well; Waters, Milford, MA, USA) and Pressure+ 96 manifold (Biotage, Uppsala, Sweden). The plant hormone metabolites were separated on Kinetex EVO C_18_ column (2.6 μm, 150 mm × 2.1 mm, Phenomenex, Torrance, CA, USA). The mobile phase A contained 5 mM ammonium acetate and 2 μM medronic acid in water, while phase B consisted of 95% (*v*/*v*) acetonitrile in water. The following gradient was used: 5% B in 0 min, 5–7% B (0.1–5 min), 10–35% B (5.1–12 min), and 35–100% B (12–13 min), followed by a 1 min hold at 100% B (13–14 min) and return to 5% B. Hormone analysis was performed with an LC/MS system consisting of UHPLC 1290 Infinity II (Agilent, Santa Clara, CA, USA) coupled to 6495 Triple Quadrupole Mass Spectrometer (Agilent, Santa Clara, CA, USA), operating in multiple reaction monitoring (MRM) mode, with quantification by the isotope dilution method. Data acquisition and processing was performed with Mass Hunter software B.08 (Agilent, Santa Clara, CA, USA).

### 2.7. Statistics

The data represent mean values obtained from at least three biological replicates, with each replicate comprising a representative sample of about 10–20 plants and three technical replicates. Statistical comparisons among the treatment variants were conducted through a one-way ANOVA, followed by Holm–Sidak test, employing Sigma Plot 11.0 software, as different letters next to the values denote significant variations. Principal component analyses (PCA) of the metabolites in and bioactivities of the examined variants were performed by *prcomp* function from *stats* package [18] in R 4.3.1 [19] following logarithmic normalization of the experimental values. PCA graphs were plotted by the R package *ggbiplot* [20]. Pearson correlation coefficients of metabolites found in *N. nuda* polar/non-polar extracts versus their biological activities were also calculated in R by *cor* function from *stats* package [21].

## 3. Results

### 3.1. Volatiles

A total of 26 volatile compounds were identified in the studied *N. nuda* samples (Table 1). Among these, the most abundant were 4a-α,7-β,7a-α-nepetalactone, 1,8-cineole/eucalyptol and germacrene D, followed by caryophyllene, β-ocimene, bicyclogermacrene, β-pinene, myrcene, and humulene. Strikingly, the nepetalactone was highly abundant in in vitro shoots. In the wild-grown *N. nuda*, iridoid was detected in developing flowers but not in leaves. On the other hand, the 1,8-cineole/eucalyptol and myrcene, were significantly increased during plant growth in natural conditions. The remaining identified volatiles were highly represented in the leaves of wild-grown plants.

### 3.2. Correlation between N. nuda Phytochemicals and Biological Activities

A preceding study revealed that extracts derived from wild-grown plants had stronger antioxidant and antiviral activities than extracts from in vitro-cultivated plants, while the antibacterial potential did not strictly hinge on the cultivation conditions ([10]; Table 2). In this context, we further investigated the anti-inflammatory effect and observed an environment-dependent trend. Extracts obtained from flowers and leaves of naturally grown plants exhibited significantly greater activity than extracts sourced from plants cultivated in aseptic conditions (Table 2). The phytoconstituents in *N. nuda* were able to block the inflammation process through the classical complement pathway [22].

To correlate chemical constituents with the established bioactive potential, the overall data about metabolites and biological activities were subjected to statistical analyses (Appendix A; Figure 2, Figure 3 and Figure 4). The total phenolic compounds (polyphenols, flavonoids, anthocyanins) and reducing sugars (including monosaccharides with a reactive aldehyde group) quantified in this study and in our previous research [10] were compared among the studied *N. nuda* variants, and subsequently correlated with the respective biological activities (Figure 2). The highest total content of sugars was observed in flowers, followed by leaves, with in vitro shoots registering the lowest content (Figure 2a). The total phenolics content displayed a similar pattern, as well as anthocyanins, whereas flavonoids were mostly concentrated in leaves. Respectively, the antioxidant and antibacterial activities against Gram(+) *S. aureus* were clearly affected by the accumulation of reducing sugars, phenolics, and anthocyanins, as seen in the PCA plot (Figure 2b). Conversely, the PCA suggested role of more specific compounds rather than the total contents in determining the antiviral, anti-inflammatory, and antibacterial activities against Gram(−) *K. pneumoniae*. Interestingly, on the PCA plot, the biological activities were grouped as follows: antiviral with anti-inflammatory effects; antioxidant combined with antibacterial effects against *S. aureus*; and the antibacterial action against *K. pneumoniae* remained distinctly separate (Figure 2b).

### 3.3. Compounds in Polar and Non-Polar Fractions

The GC-MS analysis of the polar fraction of *N. nuda* allowed the further identification of individual primary and secondary metabolites, such as organic acids, amino acids, alcohols, sugars, and phenolics (Figure 3a; Appendix A). Among these components, only amino acids showed higher levels in in vitro shoots. A significant predominance of organic acids (methylsuccinic acid, mesaconic acid, citramalic acid), sugars (fructose, glucose, mannose, galactose, trehalose, isomaltose), and phenolics (hydroquinone, tyrosol, 4-coumaric acid, caffeic acid) was observed in wild-grown plants. Vanillic acid, gentisic acid, and isoferulic acid were notably more abundant in the leaves of wild-grown *N. nuda*, whereas in flowers, catechollactate/danshensu, rosmarinic acid, and quinic acid predominated. The metabolites having a significant correlation with specific biological activities (Appendix A) are perceptibly depicted in Figure 3b, implying a potential interdependence. The antibacterial activity against *K. pneumoniae* was most strictly grouped with the content of the amino acid proline (P16), and to a lesser extent with other amino acids (P14, P17, P18), organic acids (P2, P8, P9, P10), sugars (P29, P35), and phenolics (P46, P49, P50). The antioxidant and antibacterial activities against *S. aureus* exhibited a close linkage with the content of organic acids (P8, P9), sugars (P28, P29, P30, P31, P33, P34, P35, P36), phenolics (P38, P39, P46, P49), and quinic acid (P50). As for the antiviral and anti-inflammatory activities, these were notably associated with organic acids (P6, P7), sugars (P32, P33), and phenolics (P38, P40, P45).

Next, we investigated the non-polar fraction of *N. nuda*, enriched with fatty acids, alkanes, and sterols (Figure 4a; Appendix A). The data indicated no significant differences but rather a slight enrichment of in vitro shoots in fatty acids (pentadecanoic acid, margaric acid, linolenic acid, oleic acid, behenic acid, methyl 2-hydroxytetracosanoate), alkanes (hexadecane, octadecane, eicosane), and sterols (β-sitosterol, α-amyrin). This trend was more pronounced when compared to leaves of wild-grown plants. In flowers, a more significant increase was detected in the content of 14-methyl-hexadecanoic acid, pentacosane, branched alkanes, oleanolic acid, and ursolic acid. When examining antibacterial activity against *K. pneumoniae*, it was mainly associated with fatty acids (NP2, NP3, NP5, NP6, NP11) and one alkane (NP16) (Figure 4b). On the other hand, the antioxidant and antibacterial activities against *S. aureus* were linked to a fatty acid (NP9), an alkane (NP19), branched alkanes (NP22, NP23, NP24) and sterols (NP27, NP28). The antiviral and anti-inflammatory activities did not display any prominent associations with the compounds present in the non-polar fraction.

### 3.4. Composition of Phytohormones

To elucidate the mechanisms governing phytochemical production in *N. nuda* shoots grown in vitro and in leaves and flowers of wild-grown plants, the content of plant hormones was investigated (Table 3 and Appendix A).

During in vitro cultivation, there was a substantial increase in cytokinin (CK) and gibberellin metabolism. When comparing these results to those of wild-grown plants, the focus was directed towards the active hormone forms that can trigger signaling cascade. The active form of gibberellin GA19 content was highest in in vitro shoots, followed by wild-grown leaves, and was least abundant in wild-grown flowers. The content of active CKs were equal across the variants. In contrast, the content of active form of ABA was notably higher in flowers and leaves than in in vitro-grown shoots. The levels of JA, IAA, and SA were also significantly elevated in wild-grown plants with particular upregulation in flowers.

## 4. Discussion

There is increasing knowledge aimed at interlinking the major functional groups of plant metabolites: the primary metabolites considered as directly involved in normal growth, development, and reproduction, evident in all plants; the secondary metabolites playing a role in plant protection; and the phytohormones coordinating the internal and external signals [10,23].

In this study, we aimed to investigate and group these metabolites by evaluating their contribution to the *N. nuda* bioactive potential. The in vitro cultivation is carried out under sterile conditions, increased humidity levels, the use of sucrose from the growth medium as a source of CO_2_ through mixotrophic assimilation, and reduced light intensity, in comparison to natural conditions [24]. We demonstrated that the vegetative growth of the in vitro *N. nuda* shoots corresponded to a significant increase in the metabolism of two specific plant hormones: cytokinins and gibberellins (Figure 5a). These hormonal changes likely play a substantial role in influencing the production of phytochemicals. To better understand these findings, a comparison was made between the results obtained from the in vitro plants and those from wild-grown plants. Notably, this comparison placed some emphasis on the “active” forms of the hormones, which are known to have the potential to initiate various signaling pathways within the plant [25].

Cytokinins, being essential phytohormones, mediate signals stimulating nutrient mobilization, cell division, and shoot growth [26]. Given these functions, it is expected that their levels would be higher in tissues undergoing active development, particularly in young tissues [26]. Our findings indicated that while the metabolic forms of the cytokinins were increased, their active form did not differ between the investigated variants. This suggests slower response and consequently a reduced growth rate during in vitro cultivation, and for the plants in nature during reproductive developmental phase. Gibberellins, another class of phytohormones, are responsible for cell division in developing seedlings, and aiding in the assimilation of vital nutrients. They also mediate the elongation of shoots under shading conditions [27,28]. Therefore, it reasonable to speculate that the accumulation of gibberellins in in vitro shoots is a reflection of the relatively lower light intensity in their controlled environment. This hypothesis is reinforced by the observation that the active cytokinin content in leaves tends to decrease significantly in shaded environments [29]. Curiously, young in vitro shoots had a high content of the monoterpene nepetalactone, which along with the diterpenoids like gibberellins, are synthesized through the involvement of the methylerythritol phosphate (MEP) pathway [5,28]. On the other hand, the antagonistic to GA, the ABA hormone, exhibited limited availability in in vitro shoots, and its level increased nearly 20-fold in wild-grown plants subjected to high dehydration. In young seedlings under optimal water and nutrient availability, the abundance of cytokinins is intricately linked with a reduction in auxin levels. This relationship fosters an enhancement in the shoot-to-root growth ratio, ultimately promoting reproductive tendencies [30].

In natural habitats, where plants are exposed to more intense light intensity and possibly to other unfavorable environmental conditions, there was an induction of auxins, ABA signaling and flavonoid biosynthesis within the shoot. Notably, we observed a peak in auxin levels accompanied by carbon allocation mostly in the developing *N. nuda* flowers. Sugars play an important role in various biosynthetic processes, functioning as signals that regulate the expression of genes involved in photosynthesis, osmolyte synthesis, and sucrose metabolism [31].

Phenolic compounds, including SA, follow a similar trend as auxins sharing a common biosynthetic pathway originating from shikimate/chorismate [32]. In ref. [10], the authors summarized that phenolic compounds are synthesized and accumulated within the flowers of wild-grown plants, aligning with the findings derived from the analysis conducted here. In *N. nuda* flowers, the content of flavones has been shown to be high [10]. Our observations indicate a similar pattern for anthocyanins, a major class of shoot flavonoids. Anthocyanin biosynthesis is effectively regulated by auxin and ABA signaling pathways [30]. In *N. nuda*, we identified the highest overall quantity of flavonoids in the leaves, which is closely tied to active ABA signaling. Interestingly, the leaves of in vitro *N. nuda* shoots exhibited a decreased content of phenolics, which are typically deposed in the cell walls as plants mature, serving as a defense mechanism against herbivore invasion and UV irradiation [33]. In ref. [34] the authors established antimicrobial potential of rosmarinic acid in *Nepeta* species. Caffeic acid has been reported to activate the phytoimmune response in in vitro cultures through phenylpropanoid production and enhanced antibacterial activity [35]. In ref. [36], a comprehensive overview was provided of catecholactate antioxidant and its antiapoptotic activities, along with its regulatory role in the inflammatory responses.

The content of amino acids and some organic acids was increased in the in vitro *N. nuda* shoots, correlating with the requirement for building blocks essential for growth and the biosynthesis of secondary metabolites [31]. Moreover, when considering the dehydration status in natural conditions as opposed to in vitro conditions, it becomes evident that during dehydration of wild-grown plants, there was a notable decline in the content of amino acids, such as serine, γ-aminobutanoic acid (GABA), and isoleucine, which has also been observed in related studies [31]. On the other hand, the increased proline content in *N. nuda* flowers might be related to proline-mediated flowering, which is believed to involve a link between redox regulation and the epigenetic control of flowering [37]. Besides serving as intermediates in carbon metabolism, organic acids are also key components in plant mechanisms to cope with nutrient deficiencies, metal tolerance, and plant–microbe interactions operating at the root–soil interphase [38]. Regulators like cytokinins, auxins, and ethylene have been identified to control plant–microbe interactions and the reciprocal exchange of substances that promote growth and activate plant immune responses [39].

The fatty acids identified in in vitro *N. nuda* shoots are in similar proportion to those found in flowers. Lipids are essential not only for the biogenesis of cell membranes and their role as signal molecules, but also as a source of carbon and energy, particularly in seeds [40]. The fatty acid composition in cell membranes is predominantly composed of palmitic, stearic, oleic, linoleic, and linolenic acids. Of these, unsaturated fatty acids like oleic, linoleic, and linolenic acids are particularly important in plant defense mechanisms [40]. It is reasonable to hypothesize that in young leaves, the fatty acids act in pathogen detection, thereby conferring antibacterial activity. In wild-grown plants, sterols, such as oleanolic and ursolic acids, appear to contribute to the antimicrobial potential, as supported by previous studies [41].

On the other hand, the phytohormones from the group of jasmonates are lipids formed from α-linolenic acid that play a distinct role as signaling molecules. They are known to promote senescence and flowering, respectively, and their accumulation in *N. nuda* is not observed in young in vitro shoots but is evident in flowers of wild-grown plants. In ref. [5], the authors presumed that plants respond to JA by initially downregulating the iridoid biosynthesis at an early stage of dehydration of *N. rtanjensis* and *N. argolica* leaves, after which iridoid biosynthesis is upregulated. The overall interdependence between fatty acids, ABA, JAs, and developmental and environmental signals continues to be a subject of investigation [42]. According to our data, only the content of 14-methyl hexadecenoic acid exhibited a marked increase in flowers. A previous study has indicated the role of this fatty acid as a potent sex attractant for beetles and potentially other insects [43]. A similar function has been discussed for the nepetalactone accumulation in flowers [44]. The enhanced nepetalactone content observed in in vitro *N. nuda* plants implies that during the early phases of vegetative development, this metabolite acts in protecting the young leaves either by deterring herbivores or by attracting their natural predators [45,46]. The content of the rest of the volatile compounds is increased in wild-grown plants exposed to elevated daytime temperatures, higher light intensity, pathogen attacks, and destruction by herbivores. Accordingly, it has been demonstrated the positive influence of temperature on the accumulation of compounds like 1,8-cineole/eucalyptol, trans-caryophyllene, and germacrene, while the impact of precipitation and insolation was negative [4]. Studies on 1,8-cineole/eucalyptol have established its allelopathic and antibacterial activities [47,48], while germacrene D and caryophyllene are known to display insecticidal properties, and act as repellents against mosquitoes, ticks, and aphids [49,50].

The wild-grown *N. nuda* plants also contain branched alkanes that are among the hydrocarbons recognized by ants [51]. In support, we observed a strong attraction of ants, particularly during the later stages of development of wild-grown *N. nuda* (Figure 1d and Appendix A). Drawing from the existing literature, a possible scenario emerges where, as herbivores attacks escalate, damaged plant cells release hydrocarbons to attract ants, which might facilitate sporadic pollination and seed dispersal, enhancing plant survival prospects. In parallel, the ants generate traces of pheromone mixtures that stimulate aphid boost, leading to the respective release of honeydew, which is a major carbohydrate source for the ants [52].

The diversity and abundance of plant metabolites are dynamically shaped by environmental pressure and the ongoing co-evolution with herbivore species [23]. These metabolites, encompassing both primary and secondary types, serve not only the plants themselves but are also harnessed by adapted herbivores for multiple purposes, some of which mirror their original functions in plants. The activities of *N. nuda* have been constantly studied and continually updated [10,53]. Although secondary metabolites typically assume the role of bioactive compounds, many studies have indicated the role of primary metabolites in plant protection and human health ([54,55,56,57,58,59,60,61,62,63,64,65,66,67,68,69,70,71,72,73,74,75,76,77,78,79]; Appendix A). In this study, the interrelation between metabolite profiles of *N. nuda* and its major bioactivities allowed the grouping of respective metabolic sets, whose potential has been pointed out by previous reports or requires further validation (Figure 5b).

## 5. Conclusions

The Lamiaceae family is known for its specialized metabolites diversity which provides fabulous resources for human health, food, and agricultural applications [80]. The comparison between plants cultivated under controlled conditions in vegetative stage and wild-grown plants in the reproductive phase allowed us to simulate developmental and environmental changes that influence metabolic composition and the respective bioactivities. The data acquired in this study regarding the phytochemical potential of *N. nuda* helped to expand the knowledge about primary and specialized metabolites synthesized by this species and facilitated correlations with their associated bioactivities. Moreover, plant hormones mediating these signals are highlighted, which enlightens the mechanism of regulation of *N. nuda* growth and development and corresponding phytochemical production. Future research strategies will encompass the evaluation of individual metabolite fractions for in vitro and in vivo biological activities. Additionally, a comprehensive investigation into the specific factors governing the mechanisms behind the biosynthesis of these target compounds will be undertaken. The forthcoming studies hold the promise of shedding light on the plant physiological state and enable the optimization of biotechnological protocols for the enhanced production of valuable phytochemicals (by ex vitro adaptation and cell cultures), as well as for standardization and quality control of plant material. Such studies would facilitate more efficient utilization of medicinal plants in practice by decreasing the negative impact on environment.

## Figures and Tables

**Figure 1 metabolites-13-01099-f001:**
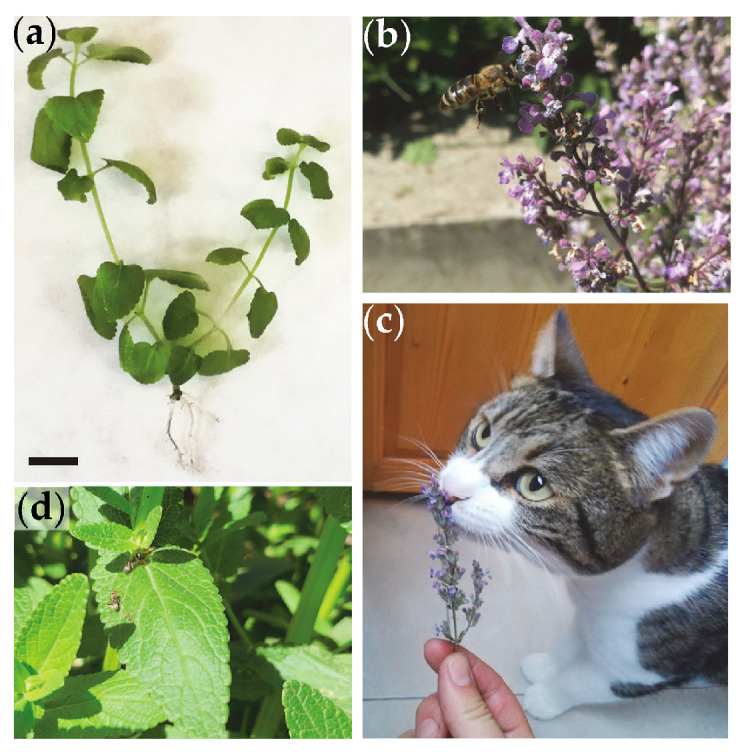
*Nepeta nuda* variants: (**a**) In vitro-cultivated plant; (**b**–**d**) Wild-grown plants that were ex vitro-adapted after in vitro cultivation: (**b**,**c**) flowers and (**d**) leaves. Scale bar: 1 cm.

**Figure 2 metabolites-13-01099-f002:**
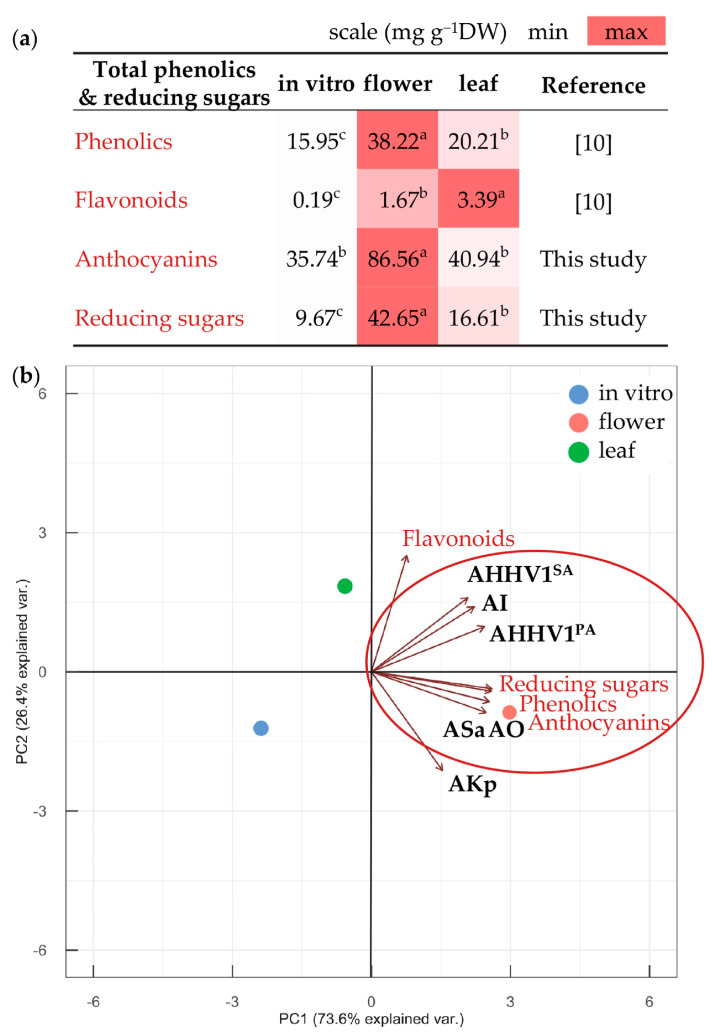
Total content of phenolics and reducing sugars in *N. nuda* samples. Analyses include shoots of in vitro-cultivated plants and flowers and leaves from wild-grown plants that were performed in our previous work [10] and in this study. (**a**) Heat map illustrates fluctuations in metabolite levels among plant variants for each group of metabolites. Statistical differences among variants were determined using one-way ANOVA (Holm–Sidak test), as different letters denote significant variations. (**b**) PCA visually presents metabolite classes and their biological activities highlighted in bold (see Table 2 for abbreviations).

**Figure 3 metabolites-13-01099-f003:**
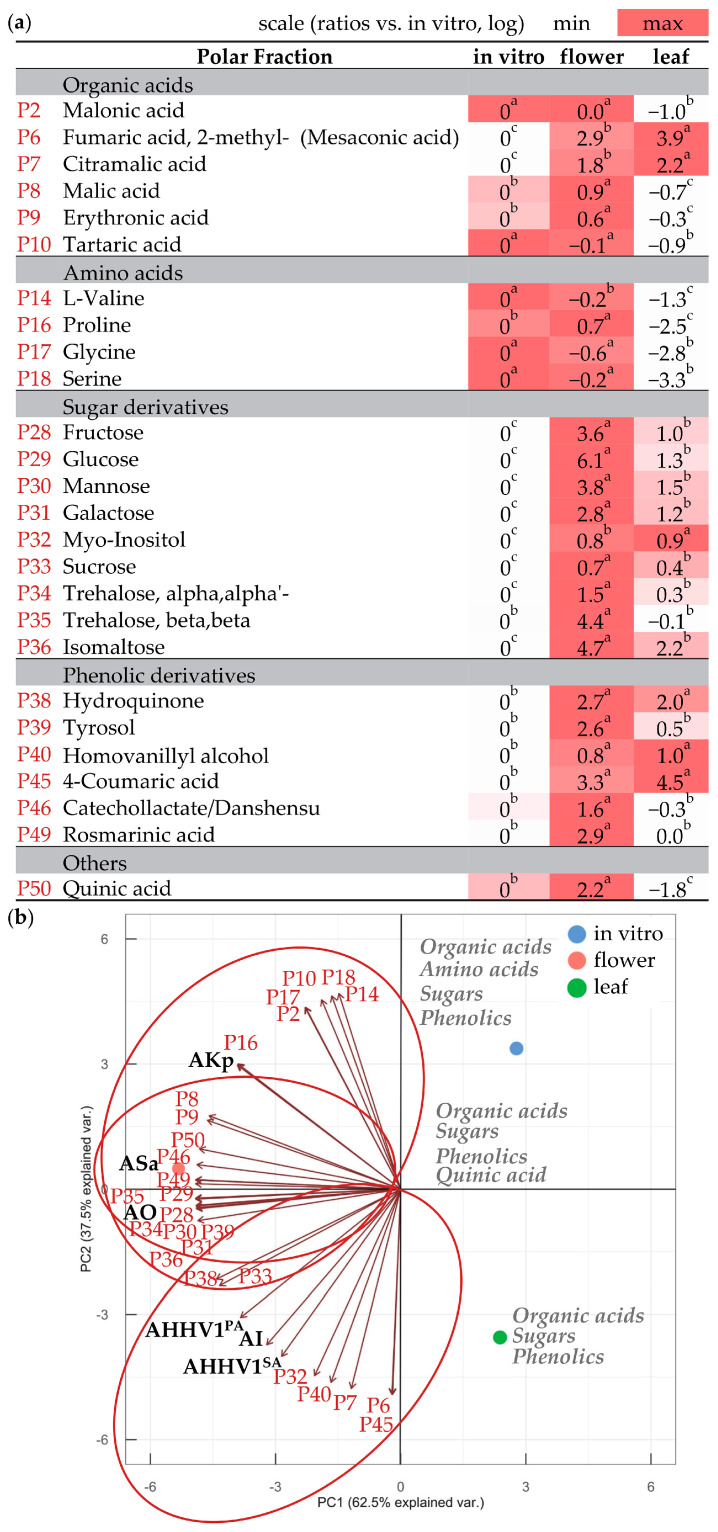
Metabolites in the polar fraction of *N. nuda* extracts that showed significant correlation with *N. nuda* biological activities. (**a**) Comparison of metabolic content of flowers and leaves of wild-grown plants with in vitro shoots, represented as relative values under logarithm. Heat map illustrates fluctuations among the plant variants for each metabolite. Statistical differences among variants were determined using one-way ANOVA (Holm–Sidak test), as different letters denote significant variations. (**b**) PCA visually presents metabolites and their biological activities highlighted in bold (see Table 2 for abbreviations; P—polar). The major metabolic groups are indicated for enhanced clarity.

**Figure 4 metabolites-13-01099-f004:**
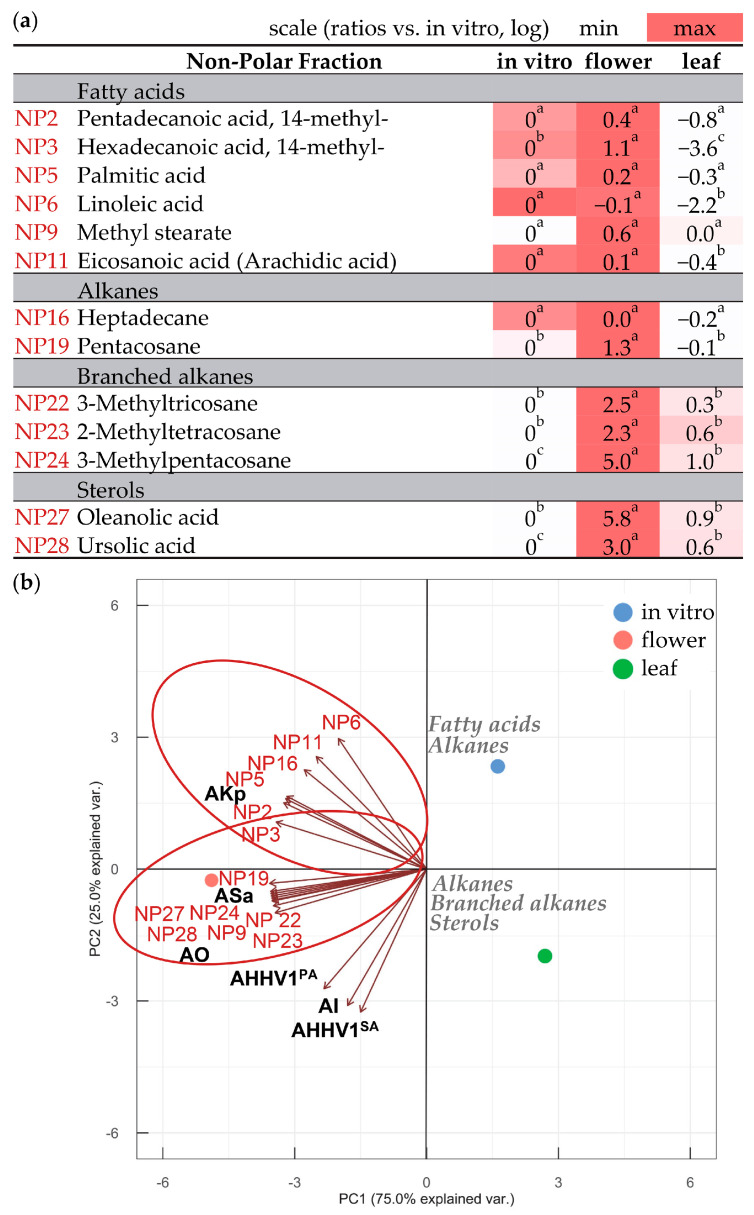
Metabolites in the non-polar fraction of *N. nuda* extracts. (**a**) Comparison of metabolic content of flowers and leaves of wild-grown plants with in vitro shoots, represented as relative values under logarithm. Heat map illustrates fluctuations among the plant variants for each metabolite. Statistical differences among variants were determined using one-way ANOVA (Holm–Sidak test), as different letters denote significant variations. (**b**) PCA visually presents metabolites and their biological activities (see Table 2 for abbreviations; NP—non-polar). The major metabolic groups are indicated for enhanced clarity.

**Figure 5 metabolites-13-01099-f005:**
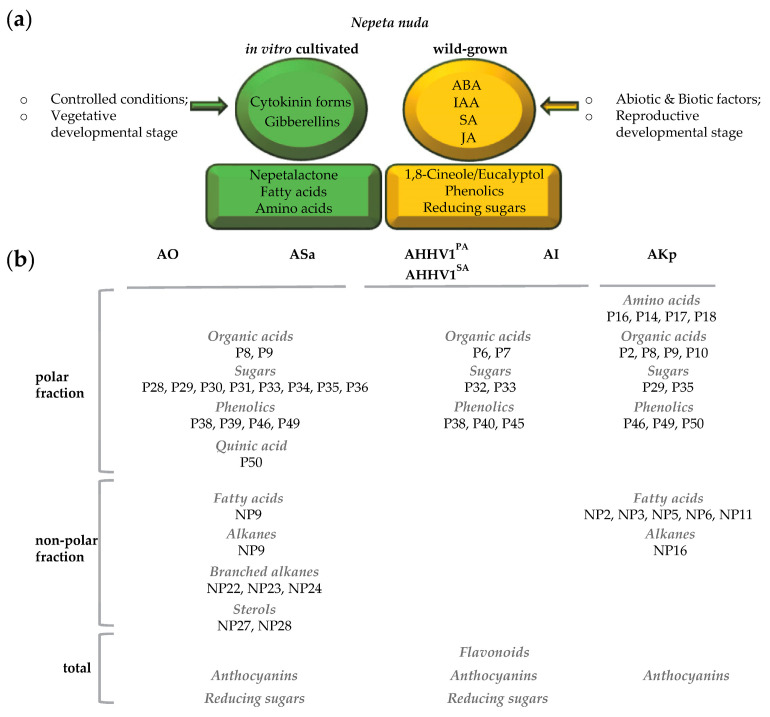
Summary of the results. (**a**) Phytohormones and metabolites under differential growth conditions. (**b**) Metabolites with putative role in biological activities. Refer to Table 2 for abbreviations.

**Table 1 metabolites-13-01099-t001:** Volatile compounds identified in *N. nuda* samples using GC/MS analysis. Analysis includes shoots of in vitro-cultivated plants and flowers and leaves from wild-grown plants. The heatmap illustrates the percentage distribution of volatiles in each sample (minimum in blue and maximum in red). Statistical differences among variants were determined using one-way ANOVA (Holm–Sidak test), as different letters denote significant variations. The most abundant compounds are displayed in bold. RT—retention time; RI—retention index.

	Volatiles	RT	RI	In Vitro	Flower	Leaf
**V1**	**β-** **Pinene**	10.18	1110.5	**0.32** **^c^**	**1.85** **^b^**	**2.22** **^a^**
V2	Sabinene	10.53	1123.4	0.15 ^c^	0.84 ^b^	0.99 ^a^
**V3**	**Myrcene**	11.75	1167.9	**0.10** **^b^**	**0.54** **^a^**	**1.16** **^a^**
V4	D-Limonene	13.01	1209.9	0.05 ^c^	0.28 ^b^	0.35 ^a^
**V5**	**1,8-Cineole/Eucalyptol**	13.34	1218.3	**3.75** **^b^**	**16.15** **^a^**	**18.44** **^a^**
V6	2-Hexenal	13.98	1230.5	0.00 ^b^	0.00 ^b^	0.84 ^а^
V7	trans-β-Ocimene	14.21	1242.2	0.06 ^b^	0.00 ^c^	0.38 ^a^
**V8**	**β-** **Ocimene**	14.87	1259.0	**0.14** **^b^**	**0.13** **^b^**	**1.79** **^a^**
V9	3-Octanone	15.19	1261.6	0.00 ^c^	0.07 ^b^	0.42 ^a^
V10	Benzene, m-di-tert-butyl-	22.03	1435.4	0.00 ^c^	0.08 ^b^	0.12 ^a^
V11	1-Octen-3-ol	22.95	1457.8	0.00 ^c^	0.27 ^b^	0.96 ^a^
V12	γ-Elemene	23.99	1489.8	0.06 ^c^	0.19 ^b^	0.65 ^a^
V13	β-Bourbonene	25.74	1527.1	0.00 ^b^	0.48 ^a^	0.46 ^a^
V14	Camphor	25.87	1530.3	0.00 ^c^	0.18 ^b^	0.32 ^a^
V15	α-Gurjunene	25.99	1539.0	0.05 ^c^	0.37 ^b^	1.34 ^a^
V16	β-Copaene	27.74	1582.4	0.25 ^c^	0.52 ^b^	0.87 ^a^
V17	β-Elemene	28.40	1598.7	0.42 ^c^	0.85 ^b^	1.52 ^a^
**V18**	**Caryophyllene**	28.71	1606.4	**2.85** **^b^**	**2.36** **^c^**	**6.07** **^a^**
**V19**	**Humulene**	31.52	1680.8	**0.56** **^b^**	**0.46** **^c^**	**1.20** **^a^**
V20	δ-Terpineol	31.67	1684.9	0.10 ^b^	0.48 ^a^	0.49 ^a^
V21	α-Terpineol	32.61	1709.7	0.25 ^b^	1.18 ^a^	1.17 ^a^
**V22**	**Germacrene D**	33.02	1720.6	**4.92** **^c^**	**7.70** **^b^**	**13.61** **^a^**
**V23**	**Bicyclogermacren**	33.95	1745.2	**0.16** **^c^**	**0.71** **^b^**	**2.32** **^a^**
V24	Benzothiazole	42.21	1977.8	0.30 ^a^	0.00 ^b^	0.00 ^b^
V25	Ledol	44.58	2042.4	0.00 ^b^	0.00 ^b^	0.54 ^a^
**V26**	**4a-** **α,7-β,7** **a-** **α-** **Nepetalactone**	45.50	2078.8	**64.41** **^a^**	**46.13** **^b^**	**0.00** **^c^**

**Table 2 metabolites-13-01099-t002:** Biological activities of *N. nuda* extracts. Analyses include shoots of in vitro-cultivated plants and flowers and leaves from wild-grown plants that were performed in our previous work [10] and in this study. Antioxidant and anti-bacterial activities were investigated in methanol extracts, and antiviral and anti-inflammatory activities—in aqueous extracts. Heat map illustrates fluctuations among the plant variants for each specific activity (maximal in red). Statistical differences among the variants were determined using one-way ANOVA (Holm–Sidak test), as different letters denote significant variations.

Bioactivities		Unit	In Vitro	Flower	Leaf	Reference
AO	Antioxidant (DPPH)	mM g^−1^DW	60.81 ^c^	206.00 ^a^	75.63 ^b^	[10]
AHHV1^SA^	Anti-human herpes virus 1 simultaneous application	% inhibition	0.00 ^b^	80.33 ^a^	75.67 ^a^	[10]
AHHV1^PA^	Anti-human herpes virus 1 postinfection application	% inhibition	0.00 ^c^	64.76 ^a^	43.52 ^b^	[10]
AI	Anti-inflammatory	% inhibition	4.64 ^c^	51.31 ^a^	44.13 ^b^	This study
ASa	Anti-bacterial (Gram+)	inhibition zone mm	10.00 ^a^	10.67 ^a^	10.00 ^a^	[10]
AKp	Anti-bacterial (Gram−)	inhibition zone mm	10.67 ^a^	12.67 ^a^	8.00 ^a^	[10]

**Table 3 metabolites-13-01099-t003:** Phytohormone content in *N. nuda* samples. Analysis includes shoots of in vitro-cultivated plants and flowers and leaves from wild-grown plants. Heat map highlights fluctuations between the plant variants for each active form of the hormone (pmol g^−1^FW; maximal in red)). Statistical differences among variants were determined using one-way ANOVA (Holm–Sidak test), as different letters denote significant variations.

Active Hormones	In Vitro	Flower	Leaf
Cytokinins (CKs)	6.76 ^a^	5.79 ^a^	6.03 ^a^
Gibberellins (GA19)	4.44 ^a^	0.62 ^b^	1.72 ^ab^
Abscisic acid (ABA)	34.59 ^b^	662.25 ^a^	696.19 ^a^
Jasmonic acid (JA)	180.64 ^c^	1773.33 ^a^	363.66 ^b^
Auxin (IAA, indole-3-acetic acid)	29.16 ^c^	305.45 ^a^	95.04 ^b^
Salicylic acid (SA)	146.22 ^c^	1020.39 ^a^	596.59 ^b^

## Data Availability

The data supporting the reported results are available and material can be provided upon request. Data is not publicly available due to privacy.

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
