# Peer review of "Uncovering the Interrelation between Metabolite Profiles and Bioactivity of In Vitro- and Wild-Grown Catmint (*Nepeta nuda* L.)"

_metabolites, 2023, doi:10.3390/metabo13101099_

Round 1

Reviewer 1 Report

Figures 2, 3 and 7 should be replaced by Table  2, 3 and 7.

In  3.2 and Figure 3. What extracts are written about?

Aqueous or methanol extracts? Write in methods.

L 270  The anti-K. pneumoniae, better antibacterial activities  against…

Add a link to Figure 8 in the results or move the figure 8 to the discussion.

Write in  2.7 what  mean -a, b, c ,  in the tables.

Author Response

Comment 1: Figures 2, 3 and 7 should be replaced by Table 2, 3 and 7.

Response: The Reviewer’s suggestion is integrated in the new manuscript version. Now, Figures 2, 3 and 7 are replaced by Table 1 [Line 256], Table 2 [Line 274] and Table 3 [Line 364].

Also, similar changes were made for the Supplementary material where now there are Table S2, Table S3 and Table S4 corresponding to the previous figures.

Comment 2: In  3.2 and Figure 3. What extracts are written about?

Aqueous or methanol extracts? Write in methods.

Response: Antioxidant and anti-bacterial activities were investigated in methanol extracts, and antiviral and anti-inflammatory activities – in aqueous extracts. The information is added to the methods. [Line 178]. Now Figure 3 is Table 2, and information is added in the Figure’s legend, as well [Line 274].

Comment 3: L 270  The anti-K. pneumoniae, better antibacterial activities  against…

Response: This change is done [now Line 319].

Comment 4: Add a link to Figure 8 in the results or move the figure 8 to the discussion.

Response: Figure 8 (now this is Figure 5) is moved to the discussion [Line 399].

Comment 5: Write in  2.7 what  mean -a, b, c ,  in the tables.

Response: This information is added [Line 237].

Reviewer 2 Report

The article is prepared very reliably, the results are presented clearly (with a small exception: Figs 3 and 4 - are the data in the tables the authors' own results or do they come from the literature, as suggested by the bibliography? Please explain and specify the table entry). The topic of the work is very interesting and current, in the era of intensive research on the phytotherapeutic potential of various plant species. A wide range of research and assessment of the possible active effects of individual groups of substances, including basic substances less frequently analyzed in this respect, are very valuable. The research used plant material from in vitro culture and organs of wild plants. It seems that the discussion and especially the conclusions lacked clear answers as to why such materials were compared and, above all, practical and final indications in this regard. Maybe it would be possible to comment on this with one or two tasks in the final part of the work? To sum up, the work is very good, minor additions do not detract from its value, I recommend it to be accepted by the publisher.

Author Response

Comment 1: The article is prepared very reliably, the results are presented clearly (with a small exception: Figs 3 and 4 - are the data in the tables the authors' own results or do they come from the literature, as suggested by the bibliography? Please explain and specify the table entry).

Response: Figure 3 is now Table 2, and Figure 4 is Figure 2. The data with Reference “Petrova et al., 2022” are our previous data. The data assigned as “This study” are from the present study. For clarity this information is added to the respective legends [Lines 275 and 302].

Comment 2: The topic of the work is very interesting and current, in the era of intensive research on the phytotherapeutic potential of various plant species.A wide range of research and assessment of the possible active effects of individual groups of substances, including basic substances less frequently analyzed in this respect, are very valuable.The research used plant material from in vitro culture and organs of wild plants.It seems that the discussion and especially the conclusions lacked clear answers as to why such materials were compared and, above all, practical and final indications in this regard.Maybe it would be possible to comment on this with one or two tasks in the final part of the work?

Response: Thanks for this comment! We added a few more information in the Conclusion part [Lines 520, 526, 534].

To sum up, the work is very good, minor additions do not detract from its value, I recommend it to be accepted by the publisher.

Reviewer 3 Report

In this work, a comparative analysis of biochemistry and bioactivity properties of catmint were analyzed depending on growth conditions. The authors have done a lot of work on the analysis of metabolite profiles. Valuable data on the composition of profiles of volatile, hydrophilic and lyophilic compounds were obtained. Differences in the accumulation of both primary and secondary metabolites between in vitro and ex vitro plants, and between leaves and flowers, were found. Differences in the biological activity of the extracts were also identified. The relationship between activity and compound profile was analyzed. Plant differences in hormone levels are also discussed in the paper. As a result, the authors draw a complex picture linking growth, hormonal status, accumulation of metabolites and biological activity.

There are still a couple of small remarks to the work.

1. The illustrative material needs improvement, it is necessary to increase the size and resolution of figures.

2.         The methods could be expanded. It is necessary to specify the age (for in vitro) and phase of plant development (beginning of flowering, transition to fruiting...). It is necessary to specify what parts of plants were sampled (certain leaves, whole shoots....).

It would be good to give references R and packages used.

In general, the work makes a good impression and provides new interesting data on the issues under study. The work may be published after few tweaks.

Author Response

In this work, a comparative analysis of biochemistry and bioactivity properties of catmint were analyzed depending on growth conditions. The authors have done a lot of work on the analysis of metabolite profiles. Valuable data on the composition of profiles of volatile, hydrophilic and lyophilic compounds were obtained. Differences in the accumulation of both primary and secondary metabolites between in vitro and ex vitro plants, and between leaves and flowers, were found. Differences in the biological activity of the extracts were also identified. The relationship between activity and compound profile was analyzed. Plant differences in hormone levels are also discussed in the paper. As a result, the authors draw a complex picture linking growth, hormonal status, accumulation of metabolites and biological activity.

There are still a couple of small remarks to the work.

Comment 1: The illustrative material needs improvement, it is necessary to increase the size and resolution of figures.

Response: In the present version, we put efforts to improve the quality of the illustrative material.

Comment 2: The methods could be expanded. It is necessary to specify the age (for in vitro) and phase of plant development (beginning of flowering, transition to fruiting...). It is necessary to specify what parts of plants were sampled (certain leaves, whole shoots....).

Response: According to the recommendations, we added more explanations about the methods. About the samples, more information is added in the 2.1. Plant Material section [Line 78], and for clarity, a supplementary figure, Figure S1, is made.

Comment 3: It would be good to give references R and packages used.

Response: This information is added, see references [18-21] [Lines 239-243].

In general, the work makes a good impression and provides new interesting data on the issues under study. The work may be published after few tweaks.